# *LINE-1* Methylation Status in Canine Splenic Hemangiosarcoma Tissue and Cell-Free DNA

**DOI:** 10.3390/ani13182987

**Published:** 2023-09-21

**Authors:** Hiroki Sato, Ken-Ichi Watanabe, Yoshiyasu Kobayashi, Mizuki Tomihari, Akiko Uemura, Michihito Tagawa

**Affiliations:** 1Veterinary Medical Center, Obihiro University of Agriculture and Veterinary Medicine, Obihiro 080-8555, Japan; 2Research Center for Global Agromedicine, Obihiro University of Agriculture and Veterinary Medicine, Obihiro 080-8555, Japan; 3Department of Veterinary Science, Osaka Metropolitan University, Izumisano 545-8585, Japan; 4Department of Veterinary Clinical Science, Obihiro University of Agriculture and Veterinary Medicine, Obihiro 080-8555, Japan; 5Department of Veterinary Associated Science, Okayama University of Science, Imabari 794-8555, Japan

**Keywords:** dog, *LINE-1*, methylation, hemangiosarcoma, cell-free DNA, liquid biopsy

## Abstract

**Simple Summary:**

In dogs, hemangiosarcoma is the most common cancer of the spleen. Although early diagnosis is effective for tumor control, benign and other malignant tumors also form splenic nodules. It is difficult to distinguish between these types of splenic tumors by imaging modalities, and splenectomy is required to confirm the diagnosis. Recently, the clinical application of cell-free DNA in plasma has been termed “liquid biopsy” and utilized as a non-invasive method for the detection of tumor-specific genetic and epigenetic alterations. *LINE-1* hypomethylation correlates well with global DNA methylation status, and it has been evaluated as a potential non-invasive tumor biomarker. This study aimed to determine the diagnostic value of *LINE-1* methylation in dogs with splenic masses using tissue and cfDNA samples. The results show that significant *LINE-1* hypomethylation was observed in hemangiosarcoma samples compared with other malignant tumors and benign groups. In addition, non-significant but similar results were observed for the cfDNA samples, thereby indicating its potential role as an early diagnostic biomarker for canine splenic hemangiosarcoma.

**Abstract:**

Splenic hemangiosarcoma is one of the most common malignant tumors in dogs, and early diagnosis is of great importance for achieving a good prognosis. DNA methylation plays an important role in cancer development. Long interspersed nuclear element 1 (*LINE-1*) is the most abundant repetitive element in the genome. *LINE-1* hypomethylation has been shown to be related to carcinogenesis in humans, and it has been used as a novel cancer biomarker. This study aimed to evaluate the methylation status of *LINE-1* in tumor tissue and circulating cell-free DNA and assess its clinical significance in canine splenic hemangiosarcoma. Genomic DNA was isolated from splenic masses of 13 dogs with hemangiosarcoma, 11 with other malignant tumors, and 15 with benign lesions. *LINE-1* methylation was quantified using methylation-sensitive and -insensitive restriction enzyme digestion followed by real-time polymerase chain reaction. Additionally, blood samples were collected from eight patients to isolate cell-free DNA to determine *LINE-1* methylation status changes during the treatment course. *LINE-1* methylation in tumor samples was significantly lower in patients with hemangiosarcoma than in those with other malignant tumors and benign lesions. Non-significant but similar results were observed for the cell-free DNA samples. Our results demonstrate that *LINE-1* methylation status is a potential biomarker for splenic hemangiosarcoma.

## 1. Introduction

Canine splenic hemangiosarcoma is the most common cancer of the spleen in dogs. Two-thirds of dogs with splenic masses reportedly have malignant tumors, with two-thirds of those tumors being hemangiosarcomas [1]. This tumor is characterized by a rapid onset of metastasis and a high propensity for spontaneous rupture, resulting in life-threatening internal hemorrhage and poor prognosis [2]. Despite surgical and medical therapies, long-term survival is rare, with the median survival time ranging from 4 to 6 months or even shorter [3]. Since the clinical stage is strongly associated with prognosis in dogs with splenic hemangiosarcoma, early diagnosis, and therapeutic intervention are important for improving outcomes [4]. Although abdominal ultrasound is recommended for the detection of splenic masses, benign lesions, and other malignant tumors also form splenic nodules, making it difficult to distinguish the type of splenic tumor using imaging techniques [5,6]. Hemangiosarcoma, other malignant tumors, and benign lesions have different prognoses; therefore, diagnostic markers that enable the early differentiation of hemangiosarcoma from these masses are required [7]. However, splenectomy for surgical resection is currently the only means to reliably confirm the histological diagnosis of tumors originating in the spleen.

Long interspersed nucleotide element-1 (*LINE-1*) constitutes about 17–18% of the human genome, and it plays a crucial role during species formation and evolution via inactivate gene functions [8,9]. *LINE-1* hypomethylation correlates well with global DNA methylation status, resulting in chromosomal instability and altered gene expression, which have been strongly correlated with carcinogenesis [9]. *LINE-1* hypomethylation has been reported in various human cancers and is associated with poor prognosis [10,11]. Circulating cell-free DNA (cfDNA) is extracellular DNA released into the bloodstream from both normal and tumor cells via apoptosis, necrosis, and secretion [12]. cfDNA contains circulating tumor DNA (ctDNA) released by cancer cells, and tumor-associated methylation changes have been detected in plasma cfDNA in numerous cancers [13]. Recently, plasma cfDNA, which reflects the hypomethylation status of repetitive DNA sequences such as *LINE-1*, has been evaluated as a potential non-invasive biomarker for the diagnosis and prognosis of various human tumors [14].

DNA methylation changes in dogs have been poorly investigated; only two reports exist regarding *LINE-1* methylation status in canine cancers [12,15]. To the best of our knowledge, no study to date has analyzed *LINE-1* methylation status in canine splenic hemangiosarcoma as a potential liquid biopsy biomarker. Therefore, this study aimed to evaluate *LINE-1* methylation status in tumor tissue and cfDNA and assess its clinical significance in canine splenic hemangiosarcoma.

## 2. Materials and Methods

### 2.1. Patient Characteristics

This retrospective study was approved by the Institutional Animal Care and Use Committee of the Obihiro University of Agriculture and Veterinary Medicine (permission numbers 20-176 and 21-123). Owner-informed consent was obtained in all cases for the participation of their animals in this study. Details on patient characteristics were extracted from medical records. Staging of the splenic tumor was determined according to the modified World Health Organization staging system [4] at the time of blood sampling and included the use of computed tomography, thoracic radiography based on three views, and abdominal ultrasound. Survival time was defined as the time from study entry to the date of death or the last follow-up.

### 2.2. Tissue Sample DNA Extraction

Genomic DNA was isolated from paraffin-embedded splenic tissues from 13 dogs with hemangiosarcomas, 11 with other malignant tumors, and 15 with benign lesions. All tumors were histologically confirmed at the Veterinary Medical Center of the Obihiro University of Agriculture and Veterinary Medicine between February 2015 and July 2020. Eight 6 μm thick slices of each tissue sample per patient were pooled and stored at −30 °C until isolation. Genomic DNA was extracted using the NucleoSpin Tissue kit (Takara Bio, Tokyo, Japan) according to the manufacturer’s instructions. DNA samples were eluted in 100 µL TE buffer and stored at −30 °C until further use. The quantity and quality of DNA were assessed using ultraviolet spectrophotometry (NanoDrop Lite, Thermo Fisher Scientific, Waltham, MA, USA).

### 2.3. Blood Collection and Cell-Free DNA Extraction

Two milliliters of peripheral blood in ethylenediaminetetraacetic acid were collected from four dogs with hemangiosarcoma, four dogs with other malignant tumors, and four dogs with benign lesions. Blood samples were collected at the initial visit without any therapeutic intervention and postoperatively in cases where monitoring was possible. The plasma was then separated via centrifugation at 2000× *g* for 10 min at 4 °C, transferred to a new tube, and centrifuged at 16,000× *g* for 10 min at 4 °C to remove cell debris. The plasma was stored at −30 °C prior to DNA extraction, and all samples were processed within 4 h of blood collection. Subsequently, cfDNA was isolated from 500 µL plasma using the MagMAX Cell-Free DNA Isolation Kit (Thermo Fisher Scientific) according to the manufacturer’s instructions. The cfDNA samples were eluted in 50 µL elution buffer and stored at −30 °C until analysis.

### 2.4. Quantification of LINE-1 Methylation

*LINE-1* methylation was measured in cf- and tissue DNA using methylation-sensitive and -insensitive restriction enzyme digestion followed by real-time polymerase chain reaction (PCR). A primer set was designed based on the sequence of the canine *LINE-1* gene (NC006621.3: 76236341–76242662 [12]), which includes three recognition sites (CCGG) for the enzymes, using Primer3Plus (https://www.primer3plus.com/, (accessed on 1 October 2020)). The set used comprised forward primer mLINE1fE (5′-AGAAACAAAGGCCTCCAAGG-3′) and reverse primer mLINE1rE (5′-GGTGCAGCTCCTGCTCCT-3′), with an expected amplicon of 109 bp (Figure 1). To determine the specificity of the primer set, PCR was performed using whole blood obtained from a healthy dog for comparison. The 20-μL PCR reaction mixture contained 10.0 μL of master mix (Promega Corporation, Madison, WI, USA), 10 pmol of each primer, 6.0 μL of distilled water, and 2.0 μL of cDNA template. Cycling conditions were as follows: initial denaturation at 95 °C for 2 min; 35 cycles of denaturation at 95 °C for 30 s; annealing at 60 °C for 30 s; extension at 72 °C for 30 s; and a final extension at 72 °C for 5 min, followed by cooling to 4 °C. Sequencing was performed at Eurofins Genomics (https://www.eurofinsgenomics.jp/, (accessed on 1 November 2020)).

One hundred nanograms of genomic DNA from tissue samples and 5 µL cfDNA were digested using 1 U of the restriction enzymes HpaⅡ and MspI (Takara Bio), respectively, in a 10 μL reaction mixture containing T buffer and 0.1% bovine serum albumin. Digestion occurred at 37 °C for 1 h followed by 80 °C heat inactivation for 20 min. Two microliters of the DNA digestion mixture were used as the template for real-time PCR. PCR was performed on a CFX Connect Real-Time PCR System (Bio-Rad, Hercules, CA, USA) using a total volume of 20 µL, which contained 500 nM of each primer, 2 µL of DNA template, and 10 µL PowerUp SYBR Green Master Mix (Thermo Fisher Scientific, Waltham, MA, USA). Initial incubation occurred at 50 °C for 2 min and 95 °C for 2 min, followed by 45 cycles, each consisting of denaturation at 95 °C for 3 s and annealing/extension at 60 °C for 30 s. A melt curve (60–95 °C) was generated at the end of each run to verify specificity. All samples were evaluated in triplicate, and a negative control (without a template) was included in each plate. The relative amount of methylated *LINE-1* was calculated using a modified 2−ΔΔ cycle threshold (CT) method: ΔCT = CT(HpaⅡ) − CT(MspI) and ΔΔCT = ΔCT(cancer) − ΔCT(healthy) [12].

### 2.5. Statistical Analyses

Statistical analyses were performed using JMP 13 software (SAS Institute, Cary, NC, USA). The Kruskal–Wallis test was used to compare the *LINE-1* methylation status in tissue and cfDNA among the groups. In addition, the *LINE-1* methylation status in tissue samples obtained from the hemangiosarcoma group and other malignant tumor groups was compared for each stage. The Steel–Dwass test was then performed on each pair of groups. Survival curves were calculated using the Kaplan–Meier method, with the medians used as cutoff values. The generalized Wilcoxon test was used to estimate survival. A value of less than 0.05 was considered statistically significant.

## 3. Results

### 3.1. Background Characteristics of the Patients

The characteristics of the 39 patients whose tissue samples were analyzed in this study are summarized in Table 1. In the other malignant tumor group, nine samples were diagnosed as sarcoma not otherwise specified, and the other two samples were diagnosed as grade Ⅲ fibrohistiocytic nodules. Of the benign lesions, seven samples were diagnosed as hematoma, four as nodular hyperplasia, three as hematoma with nodular hyperplasia, and one as myelolipoma. The median age of patients with hemangiosarcoma, other malignant tumors, and benign lesions was 11, 11, and 10 years, respectively. Plasma samples were obtained from four patients with hemangiosarcomas, three with sarcomas not otherwise specified, one with grade Ⅲ fibrohistiocytic nodule, two with hematomas, one with hematoma with nodular hyperplasia, and one with myelolipoma.

### 3.2. Comparison of LINE-1 Methylation in Tissue Samples

All DNA samples from FFPE were confirmed to have A260/280 ratios close to or greater than 1.8. The median *LINE-1* methylation levels in hemangiosarcomas, other malignant tumors, and benign lesions were 0.299 (range 0.048–0.481), 0.745 (0.311–1.019), and 0.814 (0.337–2.045), respectively. *LINE-1* methylation in tumor samples was significantly lower in patients with hemangiosarcoma than in those with other malignant tumors and benign lesions (Figure 2).

The staging was performed for 10 hemangiosarcoma and 7 other malignant tumor samples. However, there was no correlation between tumor stage and *LINE-1* methylation levels (Figure 3). In addition, the survival data of the 17 patients were evaluated after dividing them into two groups according to *LINE-1* methylation level (<0.473 vs. ≥0.473). The threshold was selected based on the median value of *LINE-1* methylation; however, there was no significant difference in survival time between the two groups (Figure 4).

### 3.3. Comparison of LINE-1 Methylation in cfDNA Samples

*LINE-1* methylation levels in cfDNA were assessed in four dogs with hemangiosarcoma, four dogs with other malignant tumors, and four dogs with benign lesions. The median *LINE-1* methylation level in dogs with hemangiosarcoma was 3.402 (2.077–4.079). On the other hand, those of the other malignant tumor and benign lesion groups were 4.740 (2.158–6.510) and 4.337 (2.345–6.497), respectively. Although no significant differences were identified between groups, *LINE-1* methylation in cfDNA tended to be lower in patients with hemangiosarcoma than in those with other malignant tumors and benign lesions (Figure 5).

### 3.4. LINE-1 Methylation Changes in cfDNA during Treatment

Blood samples were obtained approximately 2 weeks after surgery. The *LINE-1* methylation levels were assessed in three dogs with hemangiosarcoma, two dogs with other malignant tumors, and three dogs with benign lesions during the treatment course. The *LINE-1* methylation levels in cfDNA were increased in two of the three patients with hemangiosarcoma and two patients with other malignant tumors after splenectomy. In contrast, changes in methylation levels of benign lesions were mild between pre- and post-treatment (Figure 6).

## 4. Discussion

Liquid biopsy—cfDNA analysis, in particular—has emerged as a non-invasive diagnostic approach for many cancers and has recently been actively investigated. DNA methylation changes are common in various tumors, as well as in development [16]. The hypermethylation of tumor suppressor genes and/or hypomethylation of oncogenes are important events in tumorigenesis [17]. Notably, DNA methylation usually occurs in the very early stages of malignant tumors, and cfDNA methylation analysis is currently one of the most promising biomarkers for early cancer detection [18]. Several studies have investigated the role of cfDNA methylation status in cancer detection and management [19]. However, studies investigating epigenetic alterations in canine cancer are fewer than those in human cancer. Only studies regarding mammary tumors and oral melanoma have assessed cfDNA methylation status in canine cancers [12,15]. In addition, to the best of our knowledge, no previous studies have analyzed methylation status as a liquid biopsy biomarker for canine splenic hemangiosarcoma.

In this study, we first compared *LINE-1* methylation levels in tumor tissues. Methylation was significantly lower in patients with hemangiosarcoma than in those with other malignant tumors and benign lesions. In humans, genome-wide hypomethylation is associated with carcinogenesis, cancer growth and metastasis, and genomic instability in cancers [20,21]. *LINE-1* is a family of non-long terminal repeat retrotransposons, and methylation of the *LINE-1* gene is well correlated with global DNA methylation status [9]; thus, *LINE-1* is commonly used as a marker of global methylation. Several studies have revealed that *LINE-1* methylation levels are lower in cancer tissues than in normal tissues [22,23,24,25], and *LINE-1* hypomethylation was also observed in canine cancers [12,15]. Splenic hemangiosarcoma is a highly malignant tumor compared to other malignant tumors of the spleen [7]. To the best of our knowledge, this is the first report to demonstrate altered methylation levels in canine splenic hemangiosarcoma.

In human medicine, the relationship between *LINE-1* methylation level and prognosis has been investigated. *LINE-1* hypomethylation in tumor tissue correlates with clinical stage in colorectal cancer, lung tumor, and hepatocellular carcinoma [21,26,27]. However, no association was observed between *LINE-1* methylation levels in tumor tissues and the stage of canine splenic malignancies obtained in this study. In addition, there was no association between *LINE-1* methylation level and survival time. In other words, hypomethylation of *LINE-1* may be a hemangiosarcoma-specific change, and it seems to have no relationship with malignancy or prognosis in the current study. However, this study included various histological types of splenic tumors, and the treatment methods were not standardized. The hypomethylation of the *LINE-1* gene is associated with the activation of multiple proto-oncogenes and *TP53* mutations [28,29]. In particular, canine hemangiosarcomas are often associated with *TP53* mutations [30], so there may be some relationship between its hypomethylation and hemangiosarcoma tumorigenesis and malignant transformation. Further studies with a larger number of cases are needed to clarify the relationship between *LINE-1* hypomethylation and the prognosis of splenic malignancies.

Recently, cfDNA circulating in the blood has demonstrated cancer-specific methylation patterns; therefore, it has been widely applied in cancer diagnosis and monitoring using liquid biopsy [12,13,14]. In this study, plasma *LINE-1* methylation levels did not differ between groups. However, interestingly, of the patients in our study that could be followed up, four out of five dogs with hemangiosarcoma and other malignant tumors had elevated *LINE-1* methylation levels postoperatively compared to preoperatively. In contrast, there were slight changes in *LINE-1* methylation levels in dogs with benign lesions between pre- and post-operation compared to other groups. In human colorectal cancer, methylation changes have been observed in stage I–III patients whose plasma samples were matched before and after surgery [31]. The change in methylation may reflect a decrease in circulating tumor genes released from the tumor due to the reduction in tumor volume, and analysis of the methylation level of cfDNA may be useful as a tumor biomarker in canine hemangiosarcoma. However, no clear difference was noted in the prognosis between patients with increased postoperative methylation levels and those with decreased methylation levels. Because the evaluation of ctDNA as a tumor biomarker associated with canine hemangiosarcoma has not been previously reported, larger cohort studies are needed to validate these findings.

Bisulfite sequencing is the gold standard method for analysis of DNA methylation. Because ctDNA derived from malignant tumors exists only at low concentrations in those patients, a number of methods have been developed to allow for sensitive detection of ctDNA methylation [32]. Whole-genome bisulfite sequencing is the current standard for unbiased genome-wide DNA methylation profiling. However, this approach has a relatively low sequencing depth and is expensive [32]. Methylation-sensitive restriction enzyme-based quantitative PCR (MRSE-qPCR) is a rapid, easy, and cost-effective quantitative method [33]. This method is suitable for the quantitative analysis of DNA methylation in clinical samples with limited amounts of DNA, and several similar studies, including those in the field of veterinary medicine, have used MRSE-qPCR [12,33,34]. Although this method evaluates only a few target loci, the assessment of *LINE-1* methylation status reflects genome-wide methylation changes. Therefore, examining methylation status throughout the genome of hemangiosarcoma is necessary to clarify tumor development and malignant progression. In addition, in vitro functional studies are needed to determine how methylation levels are related to tumorigenesis in canine splenic hemangiosarcoma.

Hemangiosarcoma in humans, also known as angiosarcoma, is an aggressive cancer arising from the vascular endothelium in an organ or tissue. Angiosarcoma is very rare in humans. It has a poor prognosis and few standardized treatment options [35]. Recently, it has been reported that changes in the DNA methylation profile of angiosarcoma are significantly associated with chromosomal instability and prognosis, and these findings may aid the development of novel treatments for angiosarcoma [36]. Our data suggest that canine hemangiosarcoma may serve as a naturally occurring model for the development of novel therapeutics targeting epigenetic modifications in human angiosarcoma. A recent study indicated that mutations in specific driver genes can lead to aberrant methylation patterns [37]. Several driver mutations that may alter the methylation status of the entire genome have been reported in canine hemangiosarcoma [30]. The relationship between driver mutations and methylation in canine tumors should be further investigated.

This study has some limitations, including the small number of patients. The small sample size used for each analysis might have limited the extent to which differences between the groups could be detected. The inclusion of various tissue types in splenic masses in the other malignant tumor and benign lesion groups also complicates the analysis. Additionally, our study was retrospective, and the therapeutic methods were not unified; therefore, a future large-scale prospective study using standardized treatment is necessary for adequate evaluation.

## 5. Conclusions

Our results show that *LINE-1* hypomethylation in tumor samples was significantly higher in patients with hemangiosarcoma than in those with other malignant tumors and benign lesions using methylation-sensitive and -insensitive restriction enzyme digestion, followed by real-time PCR. Non-significant but similar results were observed for the cell-free DNA samples. *LINE-1* methylation could be a potential biomarker for splenic hemangiosarcoma; therefore, further research is required to establish the true clinical value of *LINE-1* methylation for canine hemangiosarcoma assessment and to clarify the relationship between epigenetic changes and hemangiosarcoma development. 

## Figures and Tables

**Figure 1 animals-13-02987-f001:**
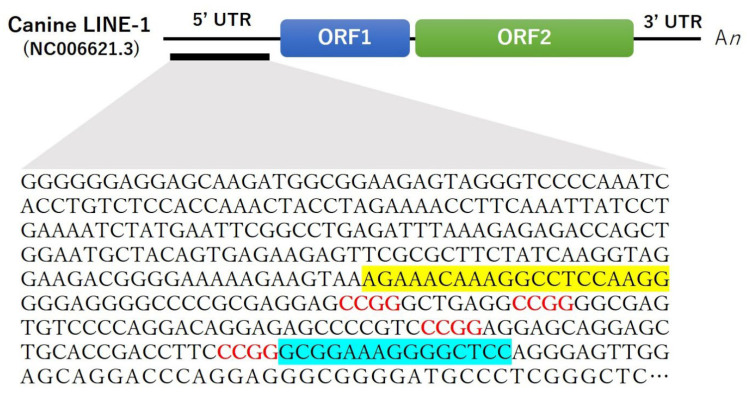
Primers for amplification of *LINE-1* for methylation-sensitive restriction enzyme digestion and real-time polymerase chain reaction. Forward and reverse primers are indicated in yellow and blue, respectively. Restriction enzyme recognition sites (CCGG) are indicated in red. *LINE-1*, long interspersed nuclear element 1; UTR, untranslated region; ORF, open reading frame.

**Figure 2 animals-13-02987-f002:**
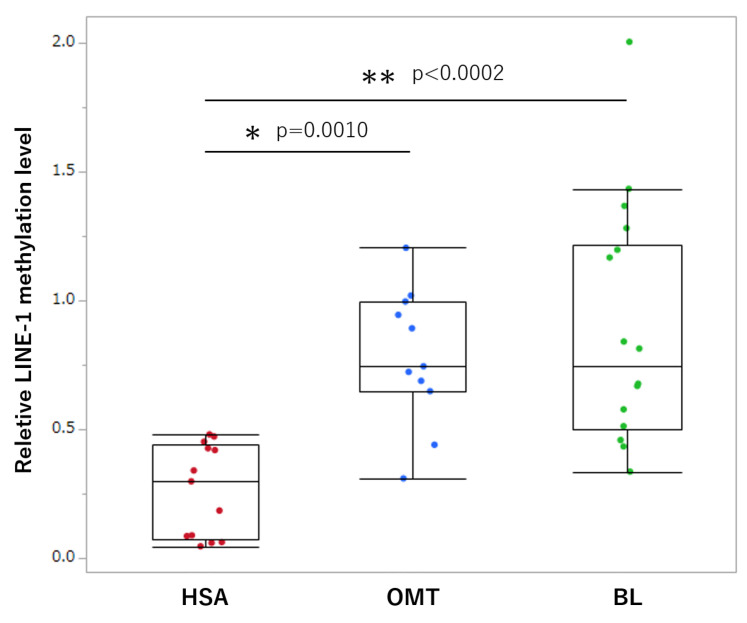
Box plots of relative *LINE-1* methylation in splenic tissue samples. Each box indicates the 25th and 75th percentiles. The horizontal line inside the box indicates the median, and the whiskers show the extreme measured values. Each dot represents a patient. HSA, hemangiosarcoma; OMT, other malignant tumors; BL, benign lesions. * *p* < 0.05; ** *p* < 0.001.

**Figure 3 animals-13-02987-f003:**
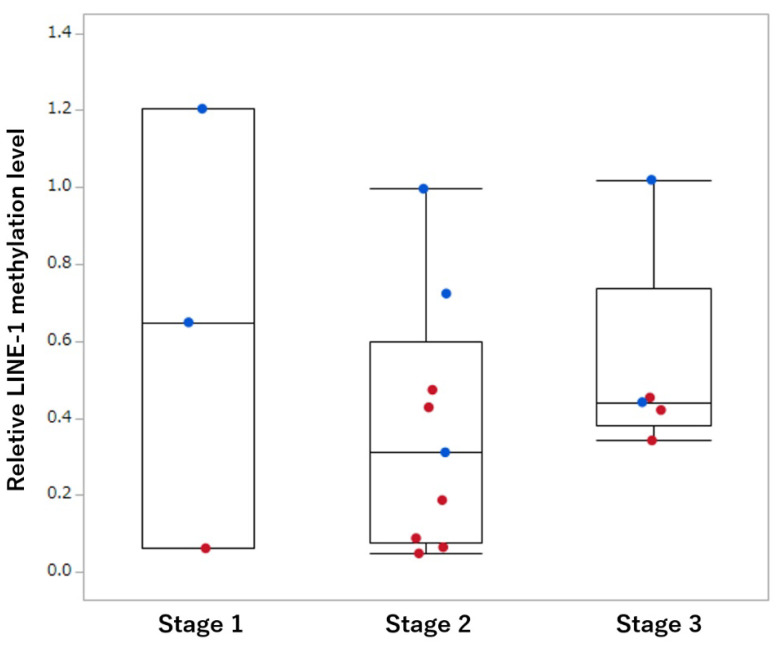
Box plot of relative *LINE-1* methylation level in malignant tumors at each stage. Each box indicates the 25th and 75th percentiles. The horizontal line inside the box represents the median, and the whiskers indicate the extreme relative methylation values. Red and blue points indicate hemangiosarcoma and other malignant tumors, respectively.

**Figure 4 animals-13-02987-f004:**
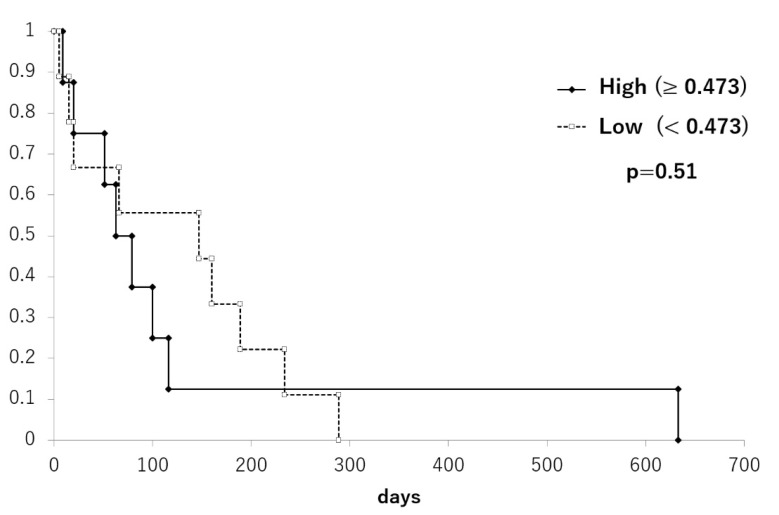
Kaplan–Meier curves of survival time in dogs with malignant tumors according to *LINE-1* methylation. Survival time was not significantly different between higher (straight line) and lower (dotted line) methylation groups.

**Figure 5 animals-13-02987-f005:**
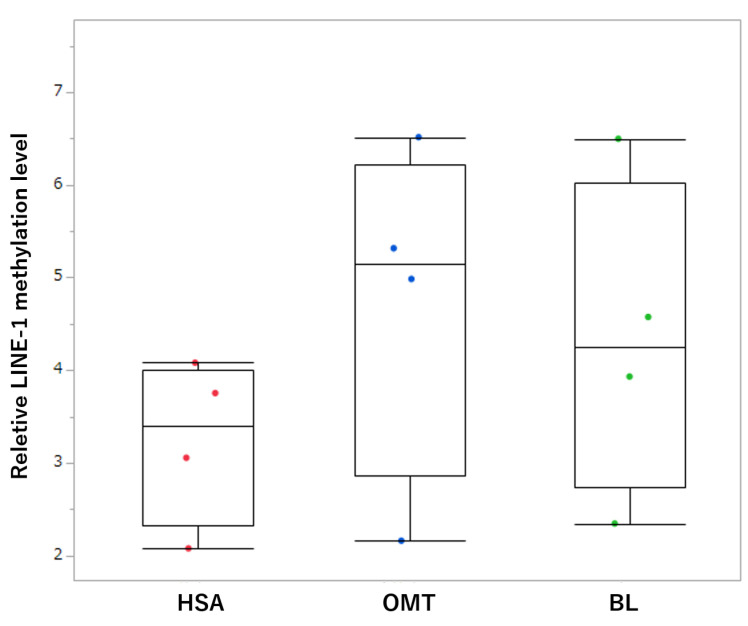
Box plot of relative *LINE-1* methylation levels in plasma cfDNA samples. Each box represents the 25th and 75th percentiles. The horizontal line inside the box indicates the median, and the whiskers illustrate the extreme methylation values. Each dot represents a patient. HSA, hemangiosarcoma; OMT, other malignant tumors; BL, benign lesions.

**Figure 6 animals-13-02987-f006:**
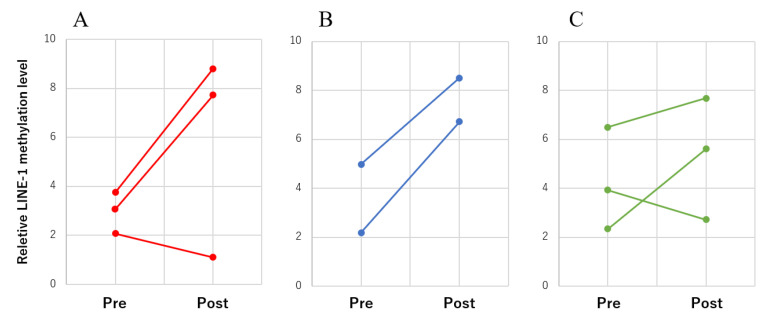
Relative *LINE-1* methylation changes in hemangiosarcomas (**A**), other malignant tumors (**B**), and benign lesions (**C**). The points connected by each line represent the same case before (pre) and after (post) splenectomy.

**Table 1 animals-13-02987-t001:** Characteristics of the patients.

Characteristics	HSA (*n* = 13)	OMT (*n* = 11)	BL (*n* = 15)
Diagnosis (*n*)	HSA (13)	Sarcoma NOS (9) FN GIII (2)	Hematoma (7) NH (4) Hematoma with NH (3) Myelolipoma (1)
Age (years; median [range])	11 (8–15)	11 (7–14)	10 (7–13)
Sex			
Female	6	6	8
Male	6	5	7
Unknown	1	-	-
Tumor stage ^a^			
1	1	2	-
2	6	3	-
3	3	2	-
Unknown	3	4	-

^a^ According to the modified World Health Organization staging system [4]. HSA, hemangiosarcoma; OMT, other malignant tumor; BL, benign lesion; NOS, sarcoma not otherwise specified; FN GⅢ, grade Ⅲ fibrohistiocytic nodules; NH, nodular hyperplasia.

## Data Availability

The data presented in this study are available on request from the corresponding author.

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
