# Peer review of "LINE-1 Methylation Status in Canine Splenic Hemangiosarcoma Tissue and Cell-Free DNA"

_animals, 2023, doi:10.3390/ani13182987_

Round 1
Reviewer 1 Report
TITLE: LINE-1 Methylation Status in Canine Splenic Hemangiosarcoma Tissue and Cell-free DNA: a Retrospective Study
Reviewer's Title Suggestion: LINE-1 Methylation Status in Canine Splenic Hemangiosarcoma Tissue and Cell-free DNA.
The cell-free DNA segment of the study was not respective but prospective.
1) Brief Summary
This work aimed to "evaluate LINE-1 methylation status in tumor tissue and cfDNA and assess its clinical significance in canine splenic hemangiosarcoma."
To achieve their goals, the authors obtained genomic DNA from paraffin-embedded splenic tissues from 13 dogs with hemangiosarcoma, 11 with other malignant tumors, and 15 with benign lesions. Furthermore, blood was collected from four dogs with hemangiosarcoma, four with other malignant tumors, and four with benign lesions. The results indicated that LINE-1 methylation in tumor samples was significantly lower in patients with hemangiosarcoma than those with other malignant tumors and benign lesions. Also, despite not being significant, the results were alike for blood samples.
The manuscript is very well-written, and the results are very relevant. I have a few suggestions to improve the merit of the manuscript further.
2) General concept comments
I. Introduction
The Introduction is clear and appropriate.
II. Materials and Methods
The material and methods are presented clearly and adequately.
III. Results
I suggest a new formatting for Table 1. The current format is confusing, even with the small amount of data presented.
Also, the results presented in section 3.4, as depicted in Figure 6, are not as convincing as described in the text. In line 213, the authors affirm that "methylation levels in benign lesions did not change following treatment." However, in Figure 6C, the first line from top to bottom shows a slight increase from pre- to post-op, and the third line shows a conspicuous increase from just above 2 to nearly 6. The authors must explain this, since they claim that "methylation levels in benign lesions did not change following treatment."
IV. Discussion
Although the considerations in the Discussion are appropriate, I have two suggestions.
In lines 238 and 239, the authors claim, "Several studies have revealed that LINE-1 methylation levels are lower in cancer tissues than in normal tissues," based on two references (22 and 23). However, in the following sentence, the investigators inform that "there are only two reports on LINE-1 methylation status in canine cancers". This is a contradiction in which two references are sufficient to justify "several studies" and "only two reports" in subsequent sentences. I would increase the number of references related to the first sentence.
For the reasons explained above in the "III. Results" comments, I suggest that the sentence "In contrast, postoperative LINE-1 methylation levels in dogs with benign lesions did not change compared with those preoperatively" (lines 261-262) should be carefully considered, pending Figure 6C further explanation.
V. Conclusion
The authors expressed in line 312: "Similar results were observed for the cfDNA samples," referring to the observation that LINE-1 hypomethylation in tumor samples was significantly higher in patients with hemangiosarcoma. On the other hand, they were more cautious in the abstract when they stated, "Non-significant but similar results were observed for the cell-free DNA samples" (line 28). This is more appropriate, considering the results obtained.
Author Response
Response to the Reviewer 1
Thank you very much for taking the time to review this manuscript. Please find the detailed responses below and the corresponding revisions/corrections highlighted/in track changes in the re-submitted files.
TITLE: LINE-1 Methylation Status in Canine Splenic Hemangiosarcoma Tissue and Cell-free DNA: a Retrospective Study
Reviewer's Title Suggestion: LINE-1 Methylation Status in Canine Splenic Hemangiosarcoma Tissue and Cell-free DNA.
The cell-free DNA segment of the study was not respective but prospective.
Answer: I have corrected the title as you pointed out.
1) Brief Summary
This work aimed to "evaluate LINE-1 methylation status in tumor tissue and cfDNA and assess its clinical significance in canine splenic hemangiosarcoma."
To achieve their goals, the authors obtained genomic DNA from paraffin-embedded splenic tissues from 13 dogs with hemangiosarcoma, 11 with other malignant tumors, and 15 with benign lesions. Furthermore, blood was collected from four dogs with hemangiosarcoma, four with other malignant tumors, and four with benign lesions. The results indicated that LINE-1 methylation in tumor samples was significantly lower in patients with hemangiosarcoma than those with other malignant tumors and benign lesions. Also, despite not being significant, the results were alike for blood samples.
The manuscript is very well-written, and the results are very relevant. I have a few suggestions to improve the merit of the manuscript further.
2) General concept comments
- Introduction
The Introduction is clear and appropriate.
- Materials and Methods
The material and methods are presented clearly and adequately.
III. Results
I suggest a new formatting for Table 1. The current format is confusing, even with the small amount of data presented.
Answer: Table 1 was slight modified.
Also, the results presented in section 3.4, as depicted in Figure 6, are not as convincing as described in the text. In line 213, the authors affirm that "methylation levels in benign lesions did not change following treatment." However, in Figure 6C, the first line from top to bottom shows a slight increase from pre- to post-op, and the third line shows a conspicuous increase from just above 2 to nearly 6. The authors must explain this, since they claim that "methylation levels in benign lesions did not change following treatment."
Answer: The sentence was changed to “changes in methylation levels of benign lesions were mild between pre and post treatment”(p8, L224-225).
- Discussion
Although the considerations in the Discussion are appropriate, I have two suggestions.
In lines 238 and 239, the authors claim, "Several studies have revealed that LINE-1 methylation levels are lower in cancer tissues than in normal tissues," based on two references (22 and 23). However, in the following sentence, the investigators inform that "there are only two reports on LINE-1 methylation status in canine cancers". This is a contradiction in which two references are sufficient to justify "several studies" and "only two reports" in subsequent sentences. I would increase the number of references related to the first sentence.
Answer: References regarding LINE-1 hypomethylation were added [24,25] (p9, L240).
For the reasons explained above in the "III. Results" comments, I suggest that the sentence "In contrast, postoperative LINE-1 methylation levels in dogs with benign lesions did not change compared with those preoperatively" (lines 261-262) should be carefully considered, pending Figure 6C further explanation.
Answer: The sentence was changed to “there were slightly changes in LINE-1 methylation levels in dogs with benign lesions be-tween pre and post operation compared to other groups.” (p9, L277-278)
- Conclusion
The authors expressed in line 312: "Similar results were observed for the cfDNA samples," referring to the observation that LINE-1 hypomethylation in tumor samples was significantly higher in patients with hemangiosarcoma. On the other hand, they were more cautious in the abstract when they stated, "Non-significant but similar results were observed for the cell-free DNA samples" (line 28). This is more appropriate, considering the results obtained.
Answer: The sentence was changed to “Non-significant but similar results were observed for the cell-free DNA samples.”(p10,L326-327)

Reviewer 2 Report
Line 43: This sentence does not make sense, please correct. We do not evaluate splenic masses on radiographs.
Line 47: Do the authors mean tissue sample? Or splenectomy for surgical resection? Please correct and specify.
Line 53: Can the authors include 1-2 sentences in the introduction about the function of LINE-1 and the effects of global hypomethylation? The authors describe it more extensively in the discussion, however many clinicians might not be familiar with the concepts.
Line 63: ….human tumors
Line 163: Hemangiosarcoma is HSA, HS is for histiocytic sarcoma. Please correct.
Table: the table is a bit shifted, please correct.
Line 153: Can the authors describe what kind of immunohistochemistry markers were used for the diagnosis of hemangiosarcoma vs. sarcoma not otherwise specified?
Line 229: Can the authors please specify which 2 cancers?
Line 240: This sentence is unnecessary. The authors already mentioned it two times.
Line 251: Can the authors elaborate on what do they mean by hemangiosarcoma-specific changes?
Author Response
Response to the Reviewer 2
Thank you very much for taking the time to review this manuscript. Please find the detailed responses below and the corresponding revisions/corrections highlighted/in track changes in the re-submitted files.
Line 43: This sentence does not make sense, please correct. We do not evaluate splenic masses on radiographs.
Answer:“on radiographs” was deleted.
Line 47: Do the authors mean tissue sample? Or splenectomy for surgical resection? Please correct and specify.
Answer: I mean splenectomy for surgical resection because biopsies have problems with the risk of rupture and low diagnostic accuracy. So, the sentence was modified.(p2, L59-60)
Line 53: Can the authors include 1-2 sentences in the introduction about the function of LINE-1 and the effects of global hypomethylation? The authors describe it more extensively in the discussion, however many clinicians might not be familiar with the concepts.
Answer: The sentence was change to “Long interspersed nucleotide element-1 (LINE-1) constitutes about 17-18% of the hu-man genome, and its play a crucial role during species formation and evolution through inactivate gene function [8,9]. LINE-1 hypomethylation correlates well with global DNA methylation status, resulting in chromosomal instability and altered gene expression, have been strongly correlated with carcinogenesis [9].”(p2, L63-65)
Line 63: ….human tumors
Answer: “human” was added.
Line 163: Hemangiosarcoma is HSA, HS is for histiocytic sarcoma. Please correct.
Answer: HS changed to HSA.
Table: the table is a bit shifted, please correct.
Answer: The table was modified.
Line 153: Can the authors describe what kind of immunohistochemistry markers were used for the diagnosis of hemangiosarcoma vs. sarcoma not otherwise specified?
Answer: Differentiation between hemangiosarcoma and sarcoma not otherwise specified was based on morphological characteristics. The tissue samples included in this study were easy to differentiate between those tissue type.
Line 229: Can the authors please specify which 2 cancers?
Answer: “regarding mammary tumor and oral melanoma” was added. (p9, L240)
Line 240: This sentence is unnecessary. The authors already mentioned it two times.
Answer: The sentence was changed to “and LINE-1 hypomethylation was also observed in canine cancers [12,15].”(p9, L252)
Line 251: Can the authors elaborate on what do they mean by hemangiosarcoma-specific changes?
Answer: The sentence of “Hypomethylation of the LINE-1 gene is associated with the activation of multiple proto-oncogenes and TP53 mutations [28,29]. In particular, canine hemangiosarcomas are often associated with TP53 mutations [30], so there may be some relationship between its hypomethylation and hemangiosarcoma tumorigenesis and malignant transformation.” was added. (p9, L265-268)
